# Hydrogen Dynamics in Hydrated Chitosan by Quasi-Elastic Neutron Scattering

**DOI:** 10.3390/bioengineering9100599

**Published:** 2022-10-21

**Authors:** Yuki Hirota, Taiki Tominaga, Takashi Kawabata, Yukinobu Kawakita, Yasumitsu Matsuo

**Affiliations:** 1Department of Life Science, Faculty of Science & Engineering, Setsunan University, Ikeda-nakamachi, Neyagawa 572-8508, Osaka, Japan; 2Neutron Science and Technology Center, Comprehensive Research Organization for Science and Society (CROSS), Tokai, Nakagun 319-1106, Ibaraki, Japan; 3J-PARC Center, Japan Atomic Energy Agency, Tokai, Nakagun 319-1195, Ibaraki, Japan

**Keywords:** biomaterial, chitosan, hydration water dynamics, proton dynamics, proton conductivity, quasi-elastic neutron scattering (QENS)

## Abstract

Chitosan, an environmentally friendly and highly bio-producible material, is a potential proton-conducting electrolyte for use in fuel cells. Thus, to microscopically elucidate proton transport in hydrated chitosan, we employed the quasi-elastic neutron scattering (QENS) technique. QENS analysis showed that the hydration water, which was mobile even at 238 K, moved significantly more slowly than the bulk water, in addition to exhibiting jump diffusion. Furthermore, upon increasing the temperature from 238 to 283 K, the diffusion constant of water increased from 1.33 × 10^−6^ to 1.34 × 10^−5^ cm^2^/s. It was also found that a portion of the hydrogen atoms in chitosan undergo a jump-diffusion motion similar to that of the hydrogen present in water. Moreover, QENS analysis revealed that the activation energy for the jump-diffusion of hydrogen in chitosan and in the hydration water was 0.30 eV, which is close to the value of 0.38 eV obtained from the temperature-dependent proton conductivity results. Overall, it was deduced that a portion of the hydrogen atoms in chitosan dissociate and protonate the interacting hydration water, resulting in the chitosan exhibiting proton conductivity.

## 1. Introduction

Recently, significant efforts have been devoted to exploring zero-emission energy sources [1,2,3,4,5], such as hydrogen fuel, which can be considered an environmentally friendly next-generation energy source. In the majority of cases, hydrogen is converted into energy through the use of fuel cells according to the following chemical reaction: 2H_2_ + O_2_ → 2H_2_O [6]. Despite the preference for such systems due to their lack of CO_2_ emissions, a number of issues still hinder the large-scale market penetration of fuel cells. For example, the electrolytes and platinum electrodes tend to be expensive, and both electrolyte manufacture and disposal have detrimental effects on the environment.

In the context of using natural polymers in fuel applications, biomaterials are excellent candidates owing to their abundance in nature. For example, chitosan, a polysaccharide bioresource, is found in some fungi, diatoms, sponges, worms, and mollusks, and is obtained by chemically deacetylating the chitin derived from crustaceans (i.e., crabs, shrimp shells, insects, and squid pendulums) through deproteinization, decalcification, and decolorization. Although chitosan has been used in a wide range of applications in the agriculture, pharmacy, medicine, food, and textile fields [7,8,9,10], several tons of chitosan are still discarded annually. However, with the recent development of chitosan as a proton conductor, its application has expanded into energy and electrical devices due to its versatility, low cost, and environmental friendliness [11,12].

Chitosan is characterized by a highly oriented structure containing amino and hydroxyl groups, and, additionally, it is known for its hydrophilic properties [13]. Furthermore, chitosan is soluble in dilute organic acids (i.e., acetic acid, formic acid, and adipic acid) but insoluble in water, alkaline solutions, and organic solvents. Through chemical modification of the functional groups of chitosan, its properties can be optimized for various applications, including those based on low-humidity environments.

In the past decade, many review articles have focused on the application of chitosan in electrochemical devices [12,14,15]. More specifically, chitosan has been widely studied as a polymer electrolyte membrane material and as the electrode material for hydrogen–polymer electrolyte fuel cells (PEFC), direct methanol fuel cells (DMFC), and alkaline fuel cells. As previously reported, alkaline fuel cells can be constructed using composite membranes based on inorganic proton conductors [16].

Under humid conditions, chitosan is known to become a proton conductor, as in the case of other biomaterials [6,17,18,19,20,21,22,23]. Our previous studies have also shown that hydrated chitosan containing a low water content at 298K exhibits a high proton conductivity as the water content increases [24,25,26,27], which indicates that the hydration water plays an important role in proton conduction. However, in the context of proton conductivity, no microscopic investigation has been carried out into the role of water molecules bonded to the main chain. It is therefore necessary to understand the mechanism of proton conduction in chitosan from a microscopic perspective.

Recently, microscopic observations of the dynamics of hydrated polymer membranes have been reported based on the use of neutron scattering measurements [28]. For example, in the case of Nafion, a polymeric membrane bearing hydrophilic sulfonic acid groups, the slow dynamics were related to the motion of hydronium ions, while the fast dynamics were related to the motion of hydration water [28,29]. Subsequently, the dynamics of the polymers and the water molecules present in polymeric membranes were revealed over different time scales using the quasi-elastic neutron scattering (QENS) technique. The QENS method is based on a dynamic structural factor, *S* (*Q*, *ω*), a time-space diffusion correlation, and structural relaxation over a wide range of momentum transfers, *Q*, from 0.1 to several Å^−1^. This represents reciprocal space information corresponding to a characteristic length ranging from a few Å to several nm, in addition to energy transfer (*E* = ħω) from the µeV region to the meV region over the picosecond-to-nanosecond timescales [30]. These scales were matched to the distance and energy of the atomic thermal motion of the polymer and its hydration water.

Since the hydrogen atom possesses a larger neutron incoherent scattering cross-section than other atoms, the QENS analysis of hydrogen-containing materials enables us to observe the single-particle dynamics of hydrogen, such as self-diffusion, which is frequently utilized to reveal the dynamics of hydration water and structural fluctuations of the hydrated molecules [31]. Upon fitting an experimental *S* (*Q*, *ω*) with the Lorentzian function and analyzing the *Q* dependence of the obtained width and magnitude of the function using the jump-diffusion model [30], the jump length, self-diffusion coefficient, and mean residence time can be discussed. To determine the spectral contribution of the hydration water, subtraction of the spectrum of a substance hydrated with D_2_O from that of the same substance hydrated with H_2_O is frequently performed, assuming no functional differences in the substance dynamics between the D_2_O and H_2_O systems [32,33]. In this way, QENS can provide a microscopic view of the behavior of the hydrogen atoms present in fuel cell electrolytes, which can lead to an improved understanding of the properties of materials in fuel cell electrolytes, while also permitting the design optimization of next-generation fuel cells and other energy devices.

Thus, herein, we report our investigation into the proton dynamics of hydrated chitosan through the use of electrical conductivity, dielectric constant, and QENS measurements. More specifically, we employed the QENS technique to clarify for the first time the role of the chitosan hydration water from a microscopic viewpoint to promote the development of new biomaterials, such as fuel cell electrolytes, based on hydrated chitosan.

## 2. Materials and Methods

### 2.1. Sample Preparation

Figure 1 shows the preparation process employed for the chitosan samples. More specifically, the chitosan membranes were prepared using a slurry of 2 wt% chitosan nanofibers dispersed in H_2_O (BiNFi-s Chitosan) supplied by Sugino Machine Ltd. (Toyama, Japan). The chitosan contained in the nanofibers was fully deacetylated and contained no impurities, other than water. The molecular weight of the chitosan was 161.16 per monomer. The membrane was grown via suction filtration using a PTFE membrane filter. After filtration, the membrane was dried for 2 days over a phosphorus pentoxide drying agent in a desiccator to obtain the dried chitosan membranes (denoted as Dry-Chitosan). The thickness of the chitosan membrane was controlled by varying the weight of the chitosan slurry, whereby a 20 g weight of chitosan slurry was used to prepare a 0.07 mm-thick chitosan film. In addition, the Dry-Chitosan was immersed in H_2_O (Milli-Q) for 2 days to prepare the hydrated chitosan membranes (denoted “Chitosan^H^”). The Dry-Chitosan was also immersed in D_2_O (99.96% purity, Euriso-top) for 2 days to prepare a chitosan film hydrated with heavy water (denoted “Chitosan^D^”).

### 2.2. Dielectric Constant and Proton Conductivity Measurements

The chitosan membranes employed to conduct the electrical measurements were prepared to have dimensions of 2.0 cm × 2.0 cm × 0.07 mm. The dielectric constants and proton conductivities of the Dry-Chitosan and Chitosan^H^ were determined using a precision LCR meter (Agilent E4980A) in the temperature range of 230–280 K. During these measurements, the admittance, *Y*, was measured, and a parallel equivalent circuit of capacitance, *C,* and resistance, *R*′, was employed. In this case, *Y* satisfies the following Equation (1):(1)$$Y=\;\frac{1}{R^{\prime }}+j\omega^{\prime }C$$
where *ω*′ and *j* are the angular frequency and an imaginary unit, respectively. The temperature dependence of the dielectric constant was calculated directly from the imaginary part of the admittance. The relative dielectric constant of the Chitosan^H^ membrane was obtained based on the dielectric constant of a vacuum (*ε*_0_ = 8.854 × 10^−12^ (F/m)). Furthermore, the AC proton conductivity, *σ*_AC_*,* at 200 kHz was calculated from the reciprocal of *R*′.

### 2.3. QENS Experiments

The chitosan membranes used for the QENS measurements were prepared to have dimensions of 4.5 cm × 4.5 cm × 0.07 mm. Hydration of the Chitosan^H^ and Chitosan^D^ membranes was carried out to achieve 10 water molecules per chitosan monomer. These hydrated membranes were then folded into aluminum foil (42 mm × 40 mm height), rolled up, and inserted into an aluminum cylinder cell with an inner diameter of 14 mm and a wall thickness of 0.25 mm. The cell was sealed with indium wire under a He atmosphere in a glove box.

The QENS experiments were carried out using a time-of-flight near-backscattering spectrometer [34,35,36], installed at beamline BL02 (DNA) at the Materials and Life Science Experimental Facility (MLF) at the Japan Proton Accelerator Research Complex (J-PARC), Tokai, Ibaraki, Japan. During the QENS measurements, high-intensity mode was used to study the water fluctuations. More specifically, fluctuations in the hydrated structure and the hydration water were examined. For the purpose of these measurements, the energy resolution of the Si111 analyzer was set at 12 μeV, the covered energy transfer, *ћω*, ranged over the high-intensity mode from −500 to 1500 μeV, the covered momentum transfer, *Q*, ranged from 0.125 to 1.875 Å^−1^, and the detector efficiency was calibrated by the measurement of a vanadium standard. The vanadium spectra were also used to represent the resolution function, *R* (*Q*, *ω*), of the instrument. The QENS spectra were obtained at temperatures of 238, 253, 268, and 283 K using a sample changer and the position-encoded automatic cell elevator (PEACE) [37]. The QENS spectra were analyzed using DAVE [38], where the instrumental resolution was obtained from the vanadium standard.

Since the chitosan monomer contains 11 hydrogen atoms and the incoherent scattering cross-section of hydrogen is large (i.e., 80.26 barns, 10^−24^ m^2^) [31], the incoherence is ~10 times larger than the coherence, even for samples hydrated with D_2_O. Therefore, the QENS spectrum of Chitosan^D^, which is denoted as “*S*_D_ (*Q*, *ω*)”, was interpreted in terms of the single-particle dynamics of the hydrogen atoms attached to chitosan in the first approximation. On the other hand, the spectrum of Chitosan^H^, denoted “*S*_H_ (*Q*, *ω*)”, provides information associated with the hydration water and the hydrogen atoms of chitosan. The hydration water dynamics in the hydrated chitosan were singled out by subtracting *S*_D_ (*Q*, *ω*) from *S*_H_ (*Q*, *ω*) to give *S*_H−D_ (*Q*, *ω*).

## 3. Results

### 3.1. Temperature Dependences of the Relative Permittivity and the Ionic Conduction

Figure 2 shows the relationship between the AC ionic (proton) conductivity, *σ*_AC_, and the temperature, *T*, for the Chitosan^H^ and Chitosan^D^ samples. As shown, the σ_AC_ of dry chitosan is extremely small (i.e., 10^−8^ S/m). However, the hydration of chitosan by H_2_O and D_2_O resulted in significantly increased *σ*_AC_ values. In addition, the log*σ*_AC_ value for Chitosan^H^ is proportional to the reciprocal of the temperature (1/*T*) between 238 and 270 K, which indicates that proton conductivity in the Chitosan^H^ specimen obeys the Arrhenius equation between these temperatures. From this result, the activation energy for proton transport was determined to be 0.38 eV over the above temperature range. Furthermore, in the case of the Chitosan^D^ specimen, the proton conductivity increased upon increasing the temperature from ~270 K. However, below this temperature, the proton conductivity was relatively constant, thereby indicating that macroscopic ionic conduction does not take place in Chitosan^D^ below 270 K.

Figure 3 shows the temperature dependence of the dielectric constants, *ε_r_,* for the Chitosan^H^, Chitosan^D^, and Dry-Chitosan samples, wherein it can be seen that above ~240 K, the *ε_r_* value of Chitosan^H^ is larger than those of the Chitosan^D^ and Dry-Chitosan specimens, and this was attributed to the effect of the hydration water in Chitosan^H^. Thus, to estimate the effect of the chitosan hydration water, the enhanced dielectric constant was examined, as shown in the inset of Figure 3. More specifically, the enhanced dielectric constant, Δ*ε_r_*, was obtained by subtracting the dielectric constant of Dry-Chitosan from that of Chitosan^H^. As expected, Δ*ε_r_* began to increase with increasing temperature beyond ~240 K, suggesting that the hydration water becomes mobile upon the application of an AC electric field at temperatures >240 K. In contrast, the *ε_r_* value of Chitosan^D^ showed a lesser degree of increase compared to that observed for Chitosan^H^. Since the AC conductivity result suggested a lack of macroscopic ion conduction below 270 K, the dielectric response can be expected to be small, thereby accounting for this observation. However, it has also been reported that the dielectric constant of confined water is extremely small [39]. More specifically, the *ε_r_* value of bulk water is ~80, while a significantly smaller value of ~2 is observed upon decreasing the thickness of the water film to 2 nm. This result indicates that the small positive deviation of *ε_r_* observed for Chitosan^D^ from that of Dry-Chitosan can be attributed to the confinement of hydration water (D_2_O) in the interstices of the neighboring chitosan layers. In contrast, the hydration water present in Chitosan^H^ exhibits an extremely large response to the AC field, which may be related to structural relaxation upon charge fluctuation during proton conduction.

### 3.2. QENS Spectra of the Hydration Water Dynamics in the Hydrated Chitosan

As explained above, *S*_H−D_ (*Q*, *ω*) is a dynamic structural factor that relates to the water dynamics of hydrated chitosan under the assumption that the chitosan dynamics are eliminated by subtracting *S*_D_ (*Q*, *ω*) from *S*_H_ (*Q*, *ω*). Figure 4 shows the QENS profiles of *S*_H−D_ (*Q*, *ω*) recorded at different temperatures where *Q* = 1.0 Å^−1^. As shown, even at 238 K, the spectrum is significantly broader than the resolution, which indicates that some of the hydration water molecules exhibit dynamic behavior. In addition, the degree of spectral broadening gradually increased upon increasing the temperature up to 268 K, and significant broadening was observed at 283 K.

### 3.3. QENS Spectra of the Chitosan Dynamics in the Hydrated Chitosan

As described previously, *S*_D_ (*Q*, *ω*) relates to the hydrogen dynamics of chitosan hydrated with D_2_O. Thus, Figure 5 shows the QENS profile of *S*_D_ (*Q*, *ω*) over different temperatures at *Q* = 1.0 Å^−1^, wherein it can be seen that the overall spectral broadening upon increasing the temperature was similar to that observed for *S*_H_*-*_D_ (*Q*, *ω*).

## 4. QENS Analysis

### 4.1. Hydration Water Dynamics in the Hydrated Chitosan

The QENS profile shown in Figure 4 consists of at least two components that contribute to the elastic peak and the wide wings. Thus, to evaluate the hydration water dynamics in the hydrated chitosan, we initially adopted the following Equation (2):*S*_H-D_ (*Q*, *ω*) = {*A*_delta_
*δ*(*ω*) + *A*_slow_
*L*(*Γ*_slow_, *ω*) } ⊗*R* (*Q*, *ω*) + BG(2)
where *δ* is a delta function representing the elastic component that is attributed to the immobile hydration water within the energy resolution, and *L* (*Γ*_slow_, *ω*) is a Lorentz function with a half-width at half maximum (HWHM), *Γ*_slow_, which represents the slow mobile hydration water, as described later. In addition, *A*_delta_ and *A*_slow_ are the magnitudes of the corresponding components, *R* (*Q*, *ω*) is the resolution function, ⊗ is the convolution operator, and BG is the instrumental background.

As shown in Figure 6, the *S*_H-D_ (*Q*, *ω*)s values at 238, 253, and 268 K were reproduced by Equation (2). However, the value of *S*_H-D_ (*Q*, *ω*) at 283 K requires an additional Lorentz function representing the fast mobile hydration water, and so the following Equation (3) was adopted:*S*_H-D_ (*Q, ω*) = {*A*_delta_
*δ*(*ω*) + *A*_slow_
*L* (*Γ*_slow_, *ω*) + *A*_fast_
*L* (*Γ*_fast_, *ω*) } ⊗*R* (*Q*, *ω*) + BG(3)
where *L* (*Γ*_fast_, *ω*) and *A*_fast_ are the Lorentz functions, where the HWHM and *Γ*_fast_ represent the fast mobile water and its magnitude, respectively.

The *Q*^2^-dependences of *Γ*_slow_ and *Γ*_fast_ at different temperatures are shown in Figure 7. As can be seen from this figure, the value of *Γ*_slow_ at 238 K gradually increases with an increasing *Q*^2^, although the increment is smaller on the high-*Q* side. In addition, *Γ*_slow_ increased upon increasing the temperature, and the overall *Q*^2^-dependence was similar. Furthermore, it was found that *Γ*_fast_ was more than three times larger than *Γ*_slow_ at 283 K. It was also found that the *Q*^2^-dependences of *Γ*_slow_ and *Γ*_fast_ fit well to the jump-diffusion model [30] as follows:(4)$$\Gamma \left(Q\right)=\frac{DQ^{2}}{1+\left(DQ^{2}\tau \right)}$$
where *D* is the self-diffusion coefficient, and *τ* is the mean residence time. In the jump-diffusion model, a diffusive particle stays at a site for *τ* and jumps to the next site with a jump distance *l* = $\sqrt{6D\tau }$. The fitting lines are indicated by the broken curves in Figure 7.

As shown in Figure 8a,b, the obtained *D* and *τ* values for the slow mobile water lie on straight lines in their corresponding Arrhenius plots, and from these results, the activation energies of *D* and *τ* for the slow mobile water were estimated to be 0.31 and 0.30 eV, respectively. Importantly, these values are close to the activation energy required for proton conduction in hydrated chitosan (i.e., 0.38 eV).

As shown in Figure 9, the value of *l* for the slow mobile water was then estimated from *D* and *τ,* and was found to be a relatively constant value of 2.4 ± 0.02 Å. In comparison, the corresponding *l* value for the fast mobile water at 283 K was determined to be 1.1 ± 0.02 Å, which is close to that of free water [33].

The coefficients of the three terms, *A*_delta_, *A*_slow_, and *A*_fast_, obtained from fitting with Equations (2) and (3), are proportional to the number of hydration water components. Thus, the fractions (*R%*) of these terms were calculated according to Equations (5)–(7) below. In addition, Figure 10 shows the fractions of the three types of hydration water, wherein it can be seen that the fraction of the slow mobile hydration water increased with an increasing temperature, and that of the fast mobile hydration water appeared at 283 K. Notably, mobile water was observed even below 273 K.
(5)$$R_{\mathrm{immobile}}=\left(\frac{A_{\mathrm{delta}}}{A_{\mathrm{delta}\;}+A_{\mathrm{slow}\;}+A_{\mathrm{fast}\;}\;}\right)\;\times \;100$$
(6)$$R_{\mathrm{slow}}=\left(\frac{A_{\mathrm{slow}}}{A_{\mathrm{delta}\;}+A_{\mathrm{slow}\;}+A_{\mathrm{fast}\;}\;}\right)\;\times \;100$$
(7)$$R_{\mathrm{fast}}=\left(\frac{A_{\mathrm{fast}}}{A_{\mathrm{delta}\;}+A_{\mathrm{slow}\;}+A_{\mathrm{fast}\;}\;}\right)\;\times \;100$$

### 4.2. Chitosan Dynamics in the Hydrated Chitosan

Due to the fact that wide wings were observed in the QENS profiles of the *S*_D_ (*Q*, *ω*) spectra in Figure 5, the *S*_D_ (*Q*, *ω*) values were fitted separately for the immobile and mobile hydrogen atoms. As shown in Figure 11, the values of *S*_D_ (*Q*, *ω*) were reproduced at all temperatures according to Equation (8):*S*_D_ (*Q*, *ω*) = {*B*_delta_
*δ*(*ω*) + *B*_hydrogen atom_
*L*(*Γ*_hydrogen atom_, *ω*) } ⊗*R* (*Q*, *ω*) + BG(8)

The *Q*^2^-dependence of *Γ*_hydrogen atom_ at different temperatures is shown in Figure 12, wherein it can be seen that upon increasing *Q*^2^, the value of *Γ*_hydrogen atom_ increases continuously, indicating that a portion of the hydrogen atoms exhibits a diffusive character. In addition, the value of *Γ*_hydrogen atom_ was well reproduced by the jump-diffusion model (Equation (4)), as shown by the broken curves in Figure 12. It should be noted here that the chitosan monomer possesses 11 hydrogen atoms, of which the four belonging to the hydroxyl and amino groups are relatively easily dissociated compared to the other hydrogen atoms belonging to the C–H bonds; such proton diffusion may be observed by QENS.

As shown in Figure 13a,b, the obtained *D* and *τ* values for the hydrogen atoms lie reasonably on the straight lines in their respective Arrhenius plots. Based on these results, the activation energies for *D* and *τ* were estimated to be 0.30 eV, which is a comparable value to that obtained for the water dynamics in hydrated chitosan.

As shown in Figure 14, the value of *l* for the mobile hydrogen atoms between 238 and 283 K was relatively constant, i.e., 2.1 ± 0.01 Å, which represents a slightly shorter length than those related to the water dynamics in hydrated chitosan.

Subsequently, the fractions of the mobile and immobile hydrogen atoms determined by a similar method to that outlined in in Equations (5)–(7) but using two delta and Lorentz contribution components are shown in Figure 15. More specifically, a small amount of mobile hydrogen is clearly present even at 238 K, although this gradually increased a fraction of 35.1% upon increasing the temperature to 283 K. It is noteworthy that this value is close to the percentage of the four dissociable hydrogen atoms among the 11 hydrogen atoms present in the chitosan monomer.

## 5. Discussion

The purpose of this study was to elucidate the relationship between the hydration water dynamics of chitosan and hydrogen dissociation from chitosan in terms of the proton conduction mechanism in chitosan hydrated by H_2_O. Thus, to determine the hydration water dynamics of chitosan, both Chitosan^H^ and Chitosan^D^ were examined. Surprisingly, the ionic conduction of Chitosan^D^ was extremely different from that of Chitosan^H^, and as mentioned in Section 3.1, this was likely due to the long-distance diffusion of protons through the hydration H_2_O but not through the hydration D_2_O. However, if the local hydrogen dynamics in the chitosan monomer are relatively similar for Chitosan^H^ and Chitosan^D^, our strategy of determining the hydration water dynamics from the analyses of *S*_H-D_ (*Q*, *ω*) may remain reasonable, since the huge contribution of hydrogen to neutron scattering allows us to neglect the contribution of the D_2_O dynamics.

The present analyses of *S*_D_ (*Q*, *ω*) revealed that a number of the hydrogen atoms in chitosan are dissociated as protons and show a diffusive character obeying the jump-diffusion model. On the other hand, the hydration water evaluated from the analyses of *S*_H-D_ (*Q*, *ω*) exhibited a slow mobile character even at low temperatures. It should be noted that the dynamics of hydrogen atoms dissociated from chitosan and the slow mobile hydration water are similar in terms of their diffusion constants, residence times, and activation energies (see Figure 8 and Figure 13), thereby suggesting that hydrogen dissociation and water motion are closely related. We therefore assumed that the dissociation of hydrogen atoms from chitosan is comparable for Chitosan^D^ and Chitosan^H^ (i.e., *S*_D_ (*Q*, *ω*) and *S*_H_ (*Q*, *ω*)).

In terms of the jump distance, a small but distinct difference between the hydration water dynamics and the proton dissociation dynamics was observed, wherein a jump distance of 2.1 Å was obtained for the hydrogen dynamics, while a slightly longer jump distance of 2.4 Å was obtained for the slow mobile water. Upon the dissociation of a hydrogen atom from chitosan, this atom may jump to a lone pair orbital of the oxygen atom from a neighboring hydration water molecule to form a hydronium ion. Since QENS detects the single-particle dynamics of hydrogen, the “slow mobile water” can be considered a proton exchange between hydronium ions and water molecules. In addition, our results indicated that hydrogen dissociation from chitosan requires the neighboring water molecules to be closer than in the case of proton exchange between hydronium ions and water molecules. This is interesting considering that both phenomena involve the protonation of water from the viewpoint of a proton receptor. Such protonation to form a hydronium ion is considered to take place via a slow jump mode with a typical characteristic time of 150 to 500 ps in a hydrated polymer electrolyte [28,29]. Furthermore, the residence time of our slow Lorentzian component was found to range from 5 to 60 ps, which is significantly faster than the slow jump mode found in the polymer electrolyte, and also faster than the intermediate situation between the slow and fast jump modes in a polymer electrolyte.

In terms of the relationship between the temperature dependence of the hydration water content (Figure 10) and that of the mobile hydrogen originating from chitosan (Figure 15), we found similar tendencies despite the fact that their magnitudes were different. As discussed above, a proton originating from chitosan can be dissociated and jump to a neighboring hydration water molecule to form a hydronium ion, and this can be followed by proton exchange between the hydronium ion and another hydration water molecule to produce further proton conduction. Based on these considerations, we made the following assumptions to determine a suitable model for this system: (1) Among the 11 hydrogen atoms present in the chitosan monomer, the four hydrogen atoms constituting the hydroxyl and amino groups are equivalently dissociable; (2) mobile hydration water assists the dissociation process; (3) at 283 K, sufficient mobile hydration water exists around all four hydrogen atoms to ensure that they all contribute to proton conduction; and (4) on average, mobile hydration water exists around dissociable hydrogen atoms. Based on these assumptions, the following Equation (9) can be derived relating the mobile hydrogen content to the hydration water content:(9)$$\frac{B_{\mathrm{mobile}}}{B_{\mathrm{mobile}}+B_{\mathrm{immobile}}}=\frac{\left(A_{\mathrm{slow}}+A_{\mathrm{fast}}\right)\;\mathrm{at}\;\mathrm{each}\;\mathrm{temperature}}{\left(A_{\mathrm{slow}}+A_{\mathrm{fast}}\right)\;\mathrm{at}\;283K}\times \frac{4}{11}$$

Thus, as shown in Figure 16, the mobile hydrogen content estimated from the hydration water content was compared with that obtained experimentally (see Figure 15). Despite the simplicity of the model, a consistent result was obtained.

As outlined above, the QENS-derived activation energy of 0.30 eV determined for both the diffusion constant and the residence time is close to that of 0.38 eV determined from the proton conduction measurements (Figure 2), despite the fact that the two measurement methods are completely different. These results suggest that the dissociation of hydrogen from chitosan and the hydration water dynamics observed by QENS (i.e., proton exchange between hydronium ions and hydration water molecules) are the origin of proton conduction in chitosan, as shown schematically in Figure 17. More specifically, the hydrogen atoms of the easily dissociable hydroxyl and amino groups of chitosan are dissociated as protons and diffuse into the nearby hydration water to form hydronium ions. Subsequently, the mobile hydration water induces proton conduction by assisting proton exchange between the hydronium ions and additional hydration water molecules. We expect that Chitosan^D^ undergoes a proton exchange process between dissociable hydrogen atoms and D_2_O, but does not exchange between hydronium ions and D_2_O and, as a result, macroscopic ion diffusion was not observed in Chitosan^D^.

## 6. Conclusions

Herein, we reported our quasi-elastic neutron scattering (QENS) investigation of the hydrogen dynamics in hydrated chitosan using chitosan samples that had been hydrated by H_2_O or D_2_O. For this purpose, a subtraction method was employed together with electrical measurements of the proton conductivity and the dielectric constant. It was found that a portion of the chitosan hydrogen atoms exhibited diffusion over the entire temperature range from 238 to 283 K. However, in terms of the hydration water dynamics, slow mobile water at 238–283 K was distinguished from fast mobile water at 283 K. In addition, the good agreement between the activation energies of the diffusion coefficients of the hydrogen atoms and the hydration water molecules suggests a strong relationship between both dynamic behaviors. Furthermore, the mobile hydrogen atom content determined experimentally corresponded well with the probability of mobile water molecules surrounding the dissociable hydroxyl and amino group hydrogen atoms of chitosan. These results reveal for the first time that the mobile hydration water present in hydrated chitosan plays an important role in assisting the dissociation of hydrogen atoms from chitosan whilst also inducing proton diffusion. Moreover, it was found that the activation energy estimated by electrical measurements of the hydrated chitosan proton conductivity was close to that estimated by QENS. Additionally, the increase in the excess dielectric constant at higher temperatures exhibited a similar trend to the temperature dependence of the mobile hydrogen atoms and the hydration water molecules. These results indicate that the local motion information obtained using the QENS technique is relevant in terms of the corresponding macroscopic information. Overall, our results demonstrated that hydrated chitosan exhibits proton conductivity at temperatures >238 K. More specifically, proton transfer from the hydroxyl and amino groups of chitosan to the surrounding hydration water molecules plays a key role in the proton conductivity, as do the slow dynamics of the hydration water. Moreover, the similar activation energies of the protonation process and the jump diffusion of hydration water strongly suggest that the hydration water assists protonation. Finally, we concluded that QENS analysis allows a qualitative and relatively quantitative analysis of the hydration water dynamics and protonation process by distinguishing them from one another using H_2_O and D_2_O hydration. These results are expected to contribute to the potential application of chitosan as an environmentally friendly proton-conducting electrolyte for use in fuel cells.

## Figures and Tables

**Figure 1 bioengineering-09-00599-f001:**
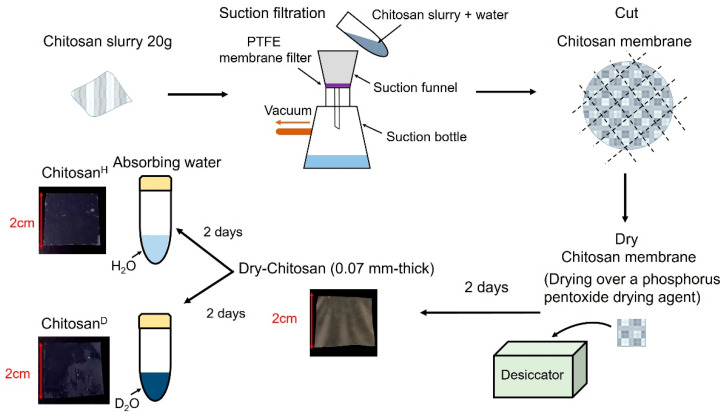
Preparation process employed to obtain the Dry-Chitosan, Chitosan^D^, and Chitosan^H^ specimens.

**Figure 2 bioengineering-09-00599-f002:**
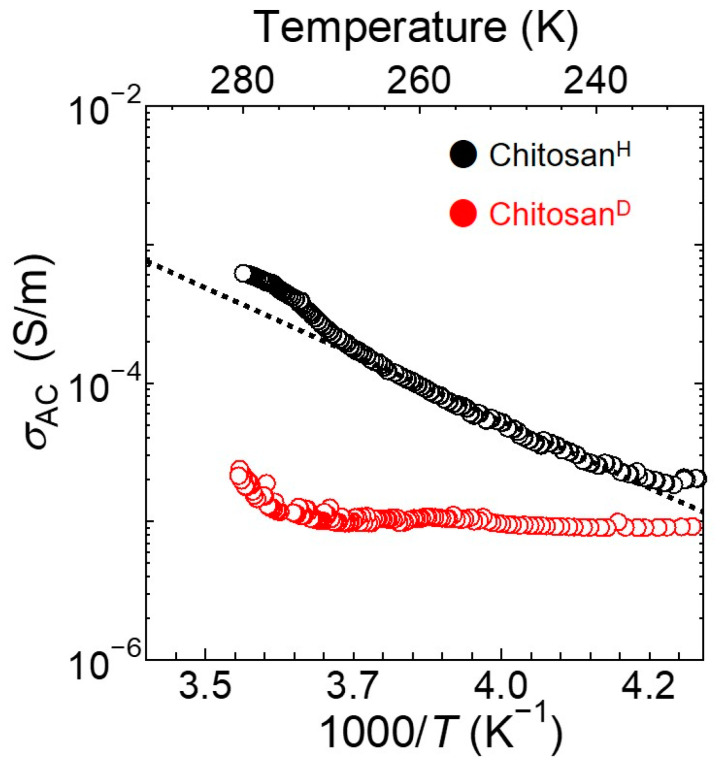
Temperature dependences of the ionic conductivities (*σ*_AC_) of Chitosan^H^ and Chitosan^D^.

**Figure 3 bioengineering-09-00599-f003:**
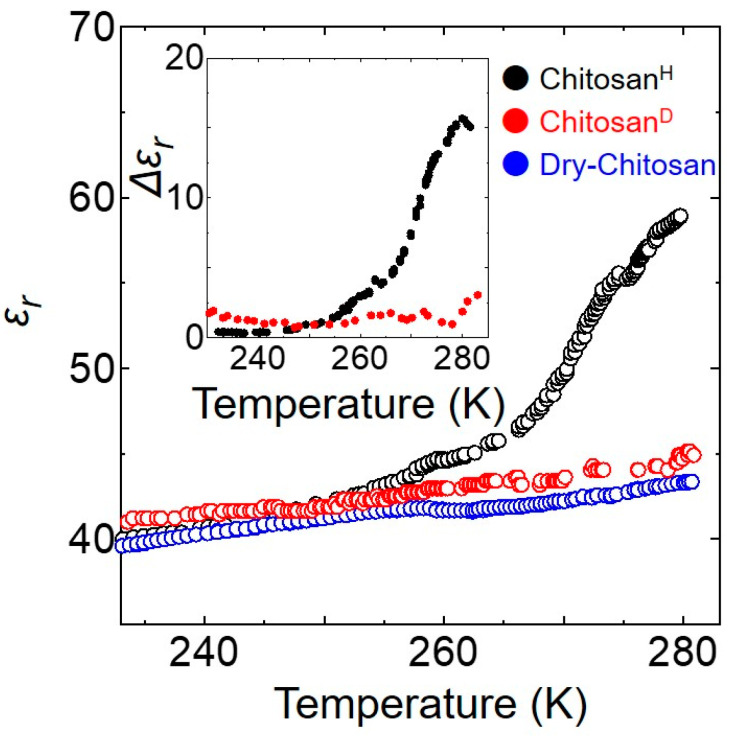
Temperature dependence of the *ε_r_* values of the Chitosan^H^, Chitosan^D^, and Dry-Chitosan samples between 230 and 280 K. The inset shows the temperature dependence of Δε*_r_*.

**Figure 4 bioengineering-09-00599-f004:**
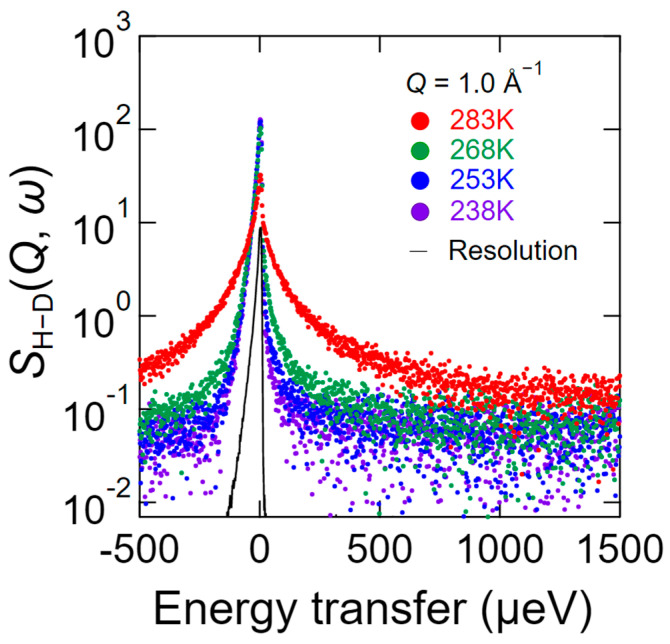
QENS profiles of *S*_H−D_ (*Q*, *ω*) at *Q* = 1.0 Å^−1^ in the energy range of −500 to 1500 μeV with a resolution of 12 μeV at 238 K (purple), 253 K (blue), 268 K (green), and 283 K (red). *S*_H−D_ (*Q*, *ω*) is the scattering intensity that represents the water dynamics of the hydrated chitosan, and is obtained by subtracting *S*_D_ (*Q*, *ω*) from *S*_H_ (*Q*, *ω*), where *S*_H_ (*Q*, *ω*) is the scattering intensity of the chitosan hydrated with H_2_O and *S*_D_ (*Q*, *ω*) is that of the chitosan hydrated with D_2_O. The black curve represents the resolution profile of vanadium obtained by QENS.

**Figure 5 bioengineering-09-00599-f005:**
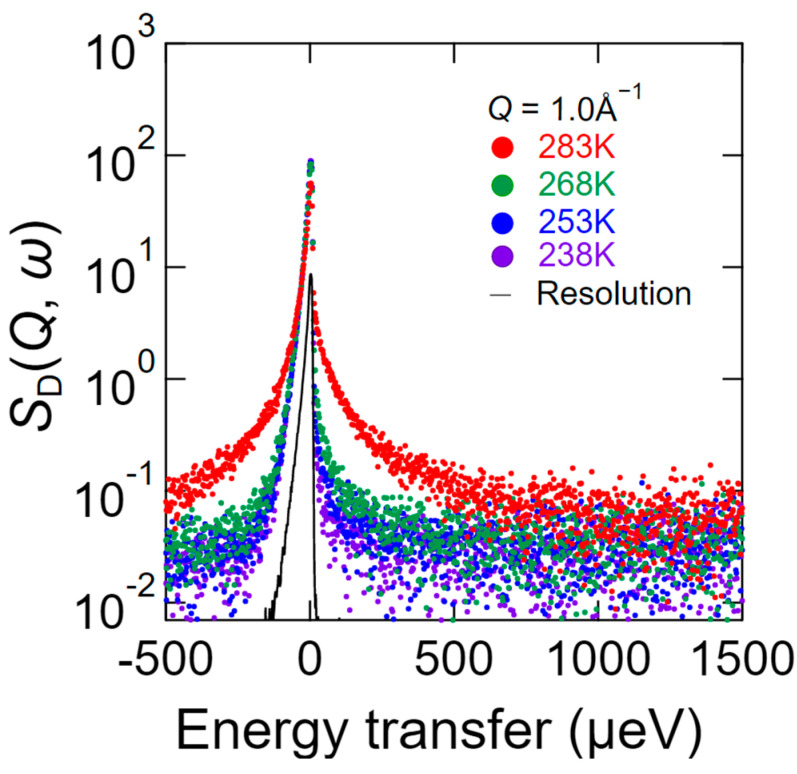
QENS profiles of *S*_D (_*Q*, *ω*) at *Q* = 1.0 Å^−1^ in the energy range of −500 to 1500 μeV with a resolution of 12 μeV at 238 K (purple), 253 K (blue), 268 K (green), and 283 K (red). *S*_D_ (*Q*, *ω*) is the scattering intensity of the chitosan hydrated with D_2_O, which is interpreted in terms of single hydrogen atom dynamics. The black curve represents the resolution profile of vanadium obtained by QENS.

**Figure 6 bioengineering-09-00599-f006:**
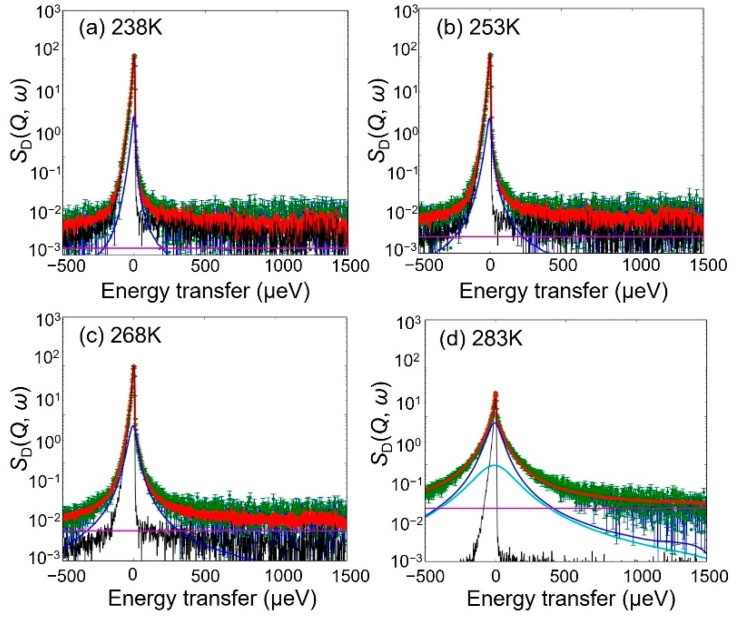
QENS spectra and fitted results for *S*_H-D_ (*Q*, *ω*) with a resolution of 12 μeV at *Q* = 1.0 Å^−1^ and at (**a**) 238 K, (**b**) 253 K, (**c**) 268 K, and (**d**) 283 K. The green points and the red lines indicate the observed intensities and the fitted curves, respectively. The black, blue, and light blue lines represent the delta, the slow Lorentz function, and the fast Lorentz function, respectively, which are convoluted by the resolution function. The magenta line represents the flat BG.

**Figure 7 bioengineering-09-00599-f007:**
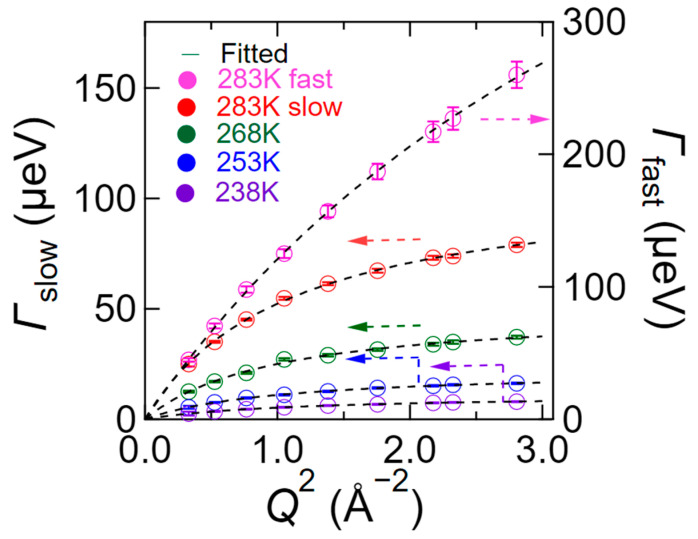
*Q*^2^-dependence of the HWHM values in *S*_H-D_ (*Q*, *ω*) obtained by profile fitting to Equations (2) and (3). The values of *Γ*_slow_ at 238 K (purple), 253 K (blue), 268 K (green), and 283 K (red) relate to the left axis, while that of *Γ*_fast_ at 283 K (pink) relates to the right axis. The broken curves of *Γ*_slow_ and *Γ*_fast_ are the fitted results obtained using the jump-diffusion model [30].

**Figure 8 bioengineering-09-00599-f008:**
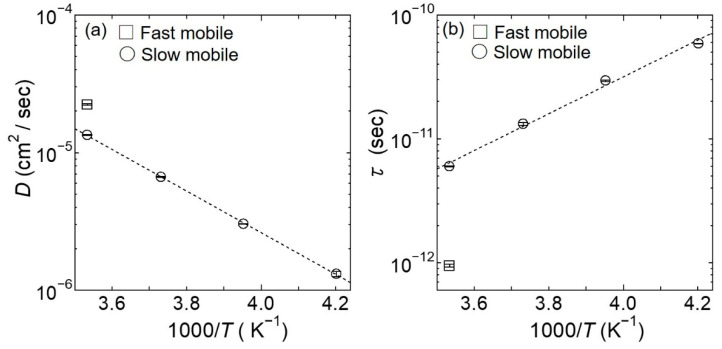
Arrhenius plots of (**a**) *D* and (**b**) *τ* for the slow mobile water at 238, 253, 268, and 283 K (depicted as circles), and for the fast mobile water at 283 K (depicted as squares), which were obtained as fitting parameters during analysis based on the jump-diffusion model for *Γ*_slow_ and *Γ*_fast_. The black dotted lines in the plots of (**a**) *D* and (**b**) *τ* for the slow mobile water represent the results fitted by the Arrhenius Equation. The errors estimated from the fitting procedure are indicated by bars.

**Figure 9 bioengineering-09-00599-f009:**
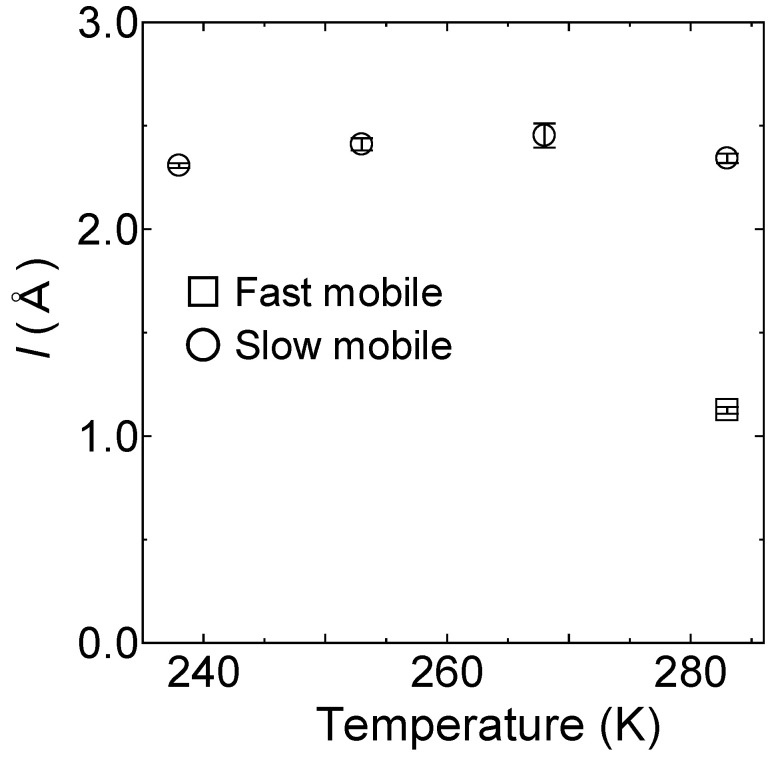
Temperature dependence of the jump distance for the slow mobile water at 238, 253, 268, and 283 K (depicted as circles), and for the fast mobile water at 283 K (depicted as squares), as obtained from the relationship *l* = $\sqrt{6D\tau }$. The errors estimated from the fitting procedure are indicated by bars.

**Figure 10 bioengineering-09-00599-f010:**
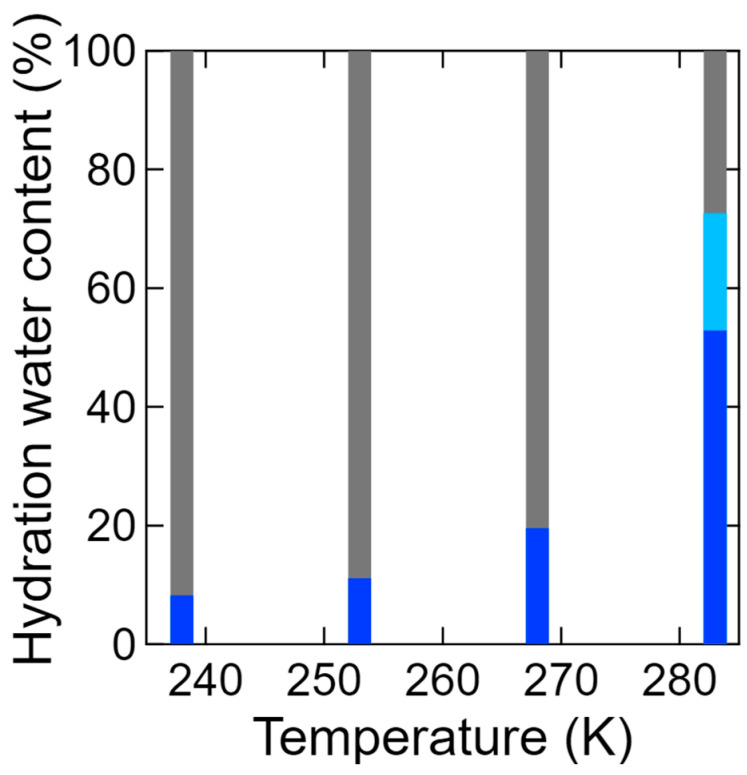
Temperature dependences of the ratios of the three types of hydration waters at 238, 253, 268, and 283 K, as determined by Equations (5)–(7). The hydration water contents for the slow and fast mobile waters and for the immobile water are denoted by blue, light blue, and gray, respectively.

**Figure 11 bioengineering-09-00599-f011:**
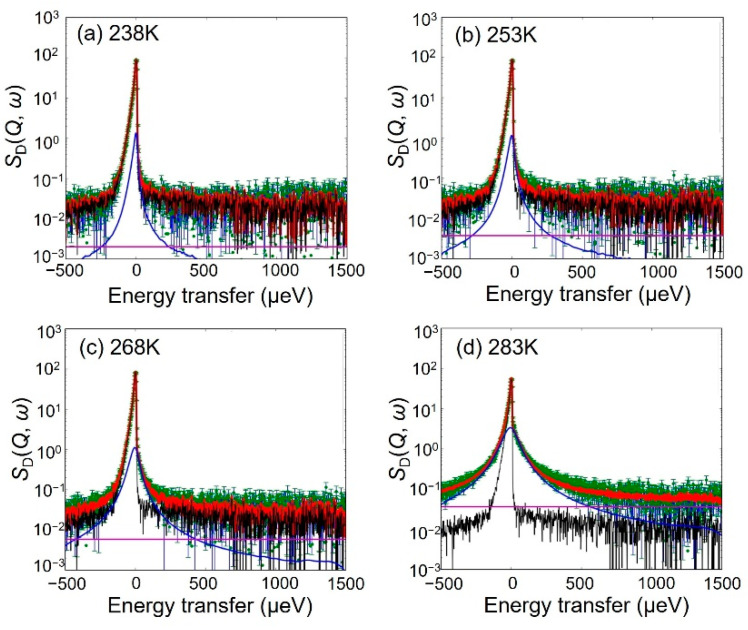
QENS spectra and fitted results of *S*_D_ (*Q*, *ω*) with a resolution of 12 μeV at *Q* = 1.0 Å^−1^ and at (**a**) 238 K, (**b**) 253 K, (**c**) 268 K, and (**d**) 283 K. The green points and red lines represent the observed intensities and the fitted curves. The black and blue lines represent the contributions from the delta and the Lorentz functions, respectively, which are convoluted by the resolution function. The magenta line represents the flat BG.

**Figure 12 bioengineering-09-00599-f012:**
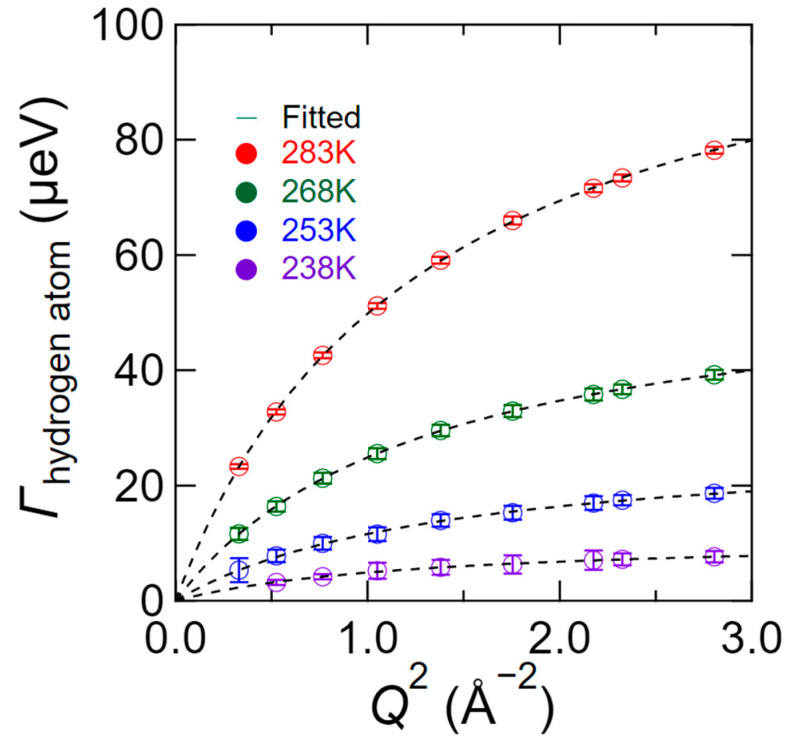
The *Q*^2^-dependence of *Γ*_hydrogen atom_ at 238 K (purple), 253 K (blue), 268 K (green), and 283 K (red) for *S*_D_ (*Q*, *ω*), as obtained by profile fitting to Equation (2). The broken curves on *Γ*_hydrogen atom_ plot represent the fitted results obtained by the jump-diffusion model [30].

**Figure 13 bioengineering-09-00599-f013:**
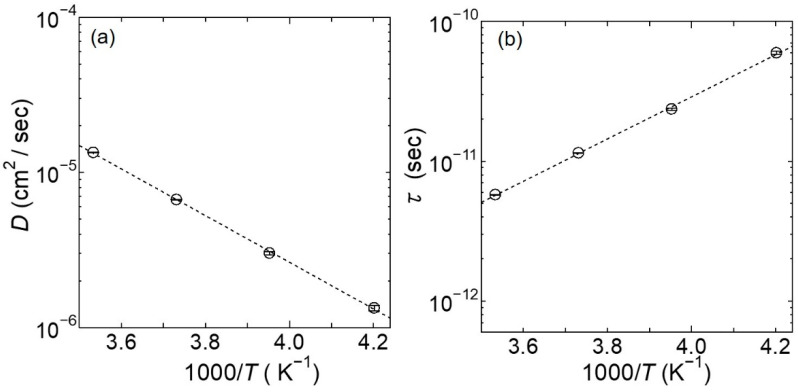
Arrhenius plots of (**a**) *D* and (**b**) *τ* for the mobile hydrogen atoms at 238, 253, 268, and 283 K obtained as fitting parameters from the analysis based on the jump-diffusion model for *Γ*_hydrogen atom_. The black dotted lines in plots (**a**,**b**) represent the results fitted by the Arrhenius Equation. The errors estimated from the fitting procedure are indicated by bars.

**Figure 14 bioengineering-09-00599-f014:**
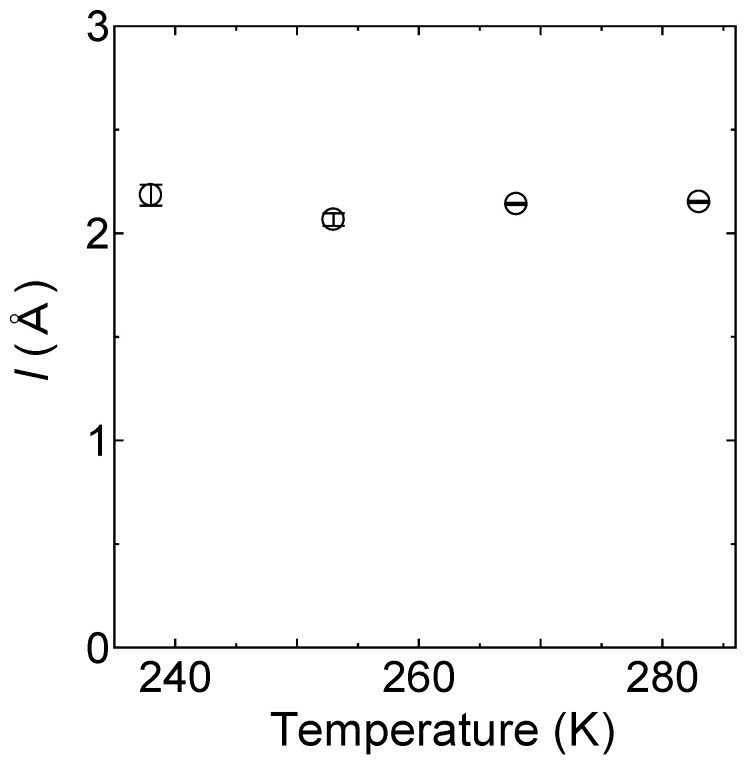
Temperature dependence of the jump distance *l* for the mobile hydrogen atoms at 238, 253, 268, and 283 K, as determined from the relationship *l* = $\sqrt{6D\tau }$. The errors estimated from the fitting procedure are indicated by bars.

**Figure 15 bioengineering-09-00599-f015:**
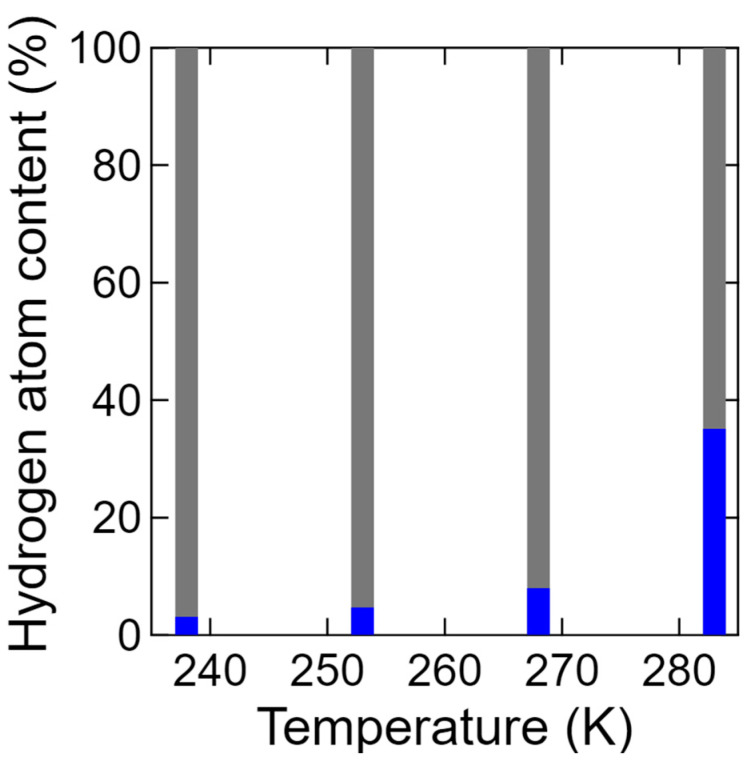
Temperature dependence of the ratio between the mobile (blue) and immobile (gray) hydrogen atoms at 238, 253, 268, and 283 K, as determined from Equations (5) and (6).

**Figure 16 bioengineering-09-00599-f016:**
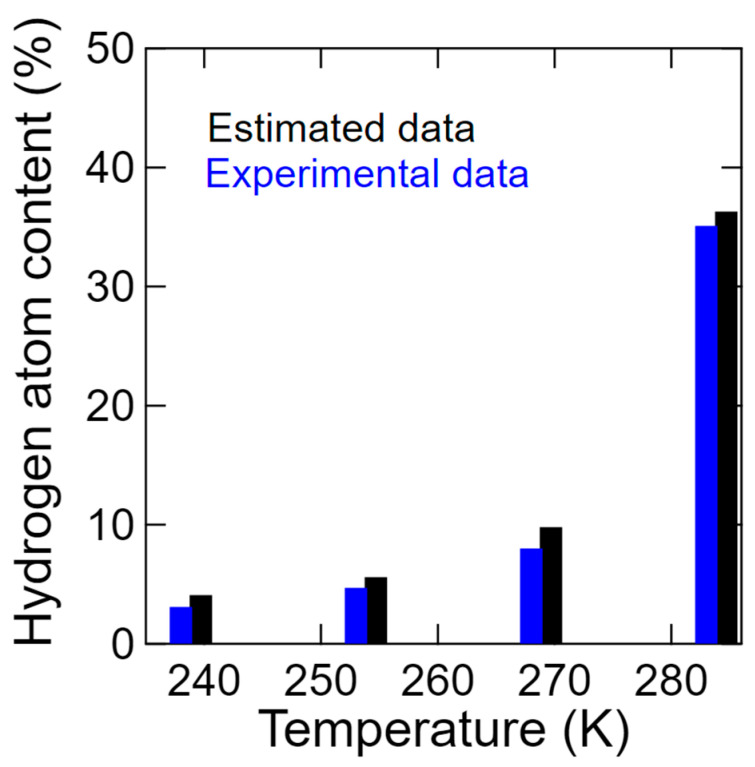
Ratios between the mobile (blue) and immobile (gray) hydrogen atoms experimentally obtained from *S*_D_ (*Q*, *ω*) and Equations (5) and (6) (blue); and those estimated using Equation (9) (black) at 238, 253, 268, and 283 K.

**Figure 17 bioengineering-09-00599-f017:**
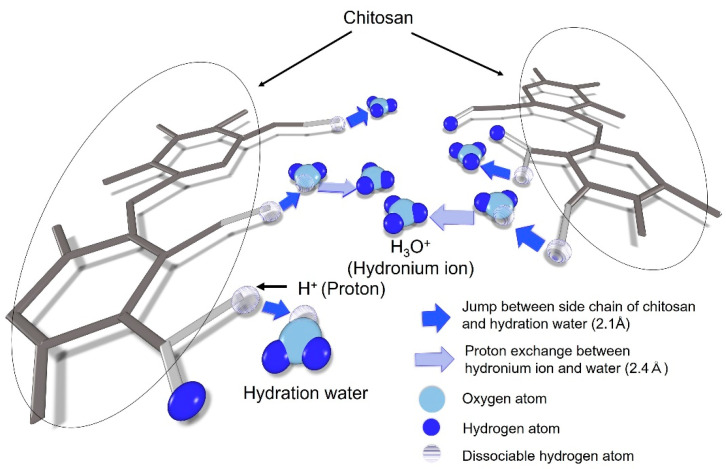
Schematic representation of the hydrogen dynamics in the vicinity of the chitosan chains in hydrated chitosan. Chitosan (dark gray), the side chains (hydroxyl and amino groups) of chitosan bearing dissociable hydrogen atoms (gray), the mobile hydration water (oxygen: light blue, hydrogen: blue), and the dissociated hydrogen atoms (translucent). The hydrogen atom of the chitosan amino group is dissociated as a proton and diffuses to a nearby hydration water molecule to form a hydronium ion. In addition, the hydrogen atoms of the hydroxyl groups are dissociated as protons and diffuse to nearby hydration water molecules followed by successive jumping to other water molecules to form hydronium ions.

## Data Availability

The datasets generated and analyzed during the current study are available from the corresponding authors upon reasonable request.

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
