# Peer review of "Hydrogen Dynamics in Hydrated Chitosan by Quasi-Elastic Neutron Scattering"

_bioengineering, 2022, doi:10.3390/bioengineering9100599_

Round 1

Reviewer 1 Report

Manuscript entitled “Hydrogen Dynamics in Hydrated Chitosan by Quasi-elastic Neutron Scattering” could be interesting for the readers. However, the paper needs a significant revision before publication. I have listed a few comments that need to be addressed:

1.       Introduction is short, add more background on the work with up-to-date citation in the introduction section.

2.       What is the importance and novelty of this work that should be clearly mentioned at the end of the introduction section?

3.       In my opinion better to include the discussion regarding the recent relevant review article on this topic in the introduction.

4.       Authors should add more details on chitosan and Quasi-elastic neutron scattering in the introduction.

5.       Write the full form once when mentioning the first instance.

6.       Authors are advised to add a schematic diagram to show the overall work for a better understanding of the readers.

7.       Discussion could be better explained. Add more discussion in this section specially authors' own insights.

8.       Improve the quality of all the Figures. Enlarge the Figure size.

9.       Use of recent citations should be preferred for the discussion of the results.

10.    The integration of the results from different parameters should be improved carefully.

11.    Conclusion could be better with key findings.

12.    Also, authors are advised to carefully revise the typos and linguistic errors to make the manuscript error-free.

Reviewer 2 Report

Specific comments:

Abstract:

 hydrated water (H2O): Is H2O a proper abbreviation for hydrated water?

which is mobile even: What do you mean "mobile even"?

Upon temperature increase: Indicate the Temp here

1. Introduction

which are easily bonded with water molecules: It is also worth mentioning the hydrophobicity of chitosan since it is soluble in slight acidic nature.

2. Materials and Methods 

2-1 sample: Start with Capital letter

purified chitosan: Provide the percentage of purity and deacetylation, and molecular weight

2 mass% chitosan nanofiber: This means 2% chitosan nanofiber? rest of 98%?

 by the weight of the chitosan slurry: So mention the actual weight of chitosan used for slurry preparation.

The hydrated chitosan membranes were prepared from the Dry-chitosan: This is a vital part, so explain the detailed procedure of making hydrated chitosan.

They are stable mechanically: Not clear, how did you check this?

un-solvable in water: In general, chitosan is not soluble in water. How did you check the solubility of hydrated chitosan membranes?

The chitosan membranes fully : Check grammar

The chitosan membranes fully hydrated by H2O: Contradictory to the previous statement (un-solvable in water). How chitosan is soluble in water? how did you prepare chitosan membrane? what is the solubility of prepared chitosan membrane?

 That done with D2O: What do you mean " That done with D2O"?

 D2O:Expand.

 is hereafter denoted as “ChitosanD: not a clear statement about ChitosanH and ChitosanD.

The dielectric constant and proton conductivity in Dry-chitosan and 90 ChitosanH: Why specifically used ChitosanH only here without ChitosanD? 

It is better to provide a Scheme about the experimental setup to understand the work more clearly.

3. Results :

Better to provide prepared Chitosan membrane pictures as Fig.

Figure 1. Temperature dependence of σAC in ChitosanH: Provide the results of ChitosanD and dry chitosan as well.

Figure 2.: Provide the result of ChitosanD.

Figures 3-15. : It is not clear which sample result is presented here. Revise the figure caption with clear sample details. Provide the results for all three samples (Dry chitosan, ChitosanH and ChitosanD).

15 Figures must be combined in an appropriate way (3-4 Figures together) together to reduce the Figure numbers. 

It is better to study the chemical and structural modifications that happened within chitosan during proton transfer.

Figure 17. : The scheme presented is not clear, especially the interaction between the chitosan side chain and water and the dissociated hydrogen atom. Revise

5. Conclusions

 chitosan hydrated by H2O or D2O: This conclusion is not supported by the present study, since some experiments were carried out with H2) and some were D2O, It is better to provide comparative data of all three H2O and D2O and dry chitosan together. 

Round 2

Reviewer 2 Report

No further comments